# An Artificial-Intelligence-Driven Spanish Poetry Classification Framework

**Shutian Deng [1], Gang Wang [2], Hongjun Wang [2,* ] and Fuliang Chang [1,*]**

[1]  School of Hispanic and Portuguese Studies, Beijing Foreign Studies University, Beijing 100089, China; bfsudst@bfsu.edu.cn
[2]  School of Computing and Artificial Intelligence, Southwest Jiaotong University, Chengdu 610031, China; 2022wg@my.swjtu.edu.cn
*   Correspondence: wanghongjun@swjtu.edu.cn (H.W.); changfuliang@bfsu.edu.cn (F.C.)

**Abstract:** Spain possesses a vast number of poems.   Most have features that mean they present significantly different styles. A superficial reading of these poems may confuse readers due to their complexity. Therefore, it is of vital importance to classify the style of the poems in advance. Currently, poetry classification studies are mostly carried out manually, which creates extremely high requirements for the professional quality of classifiers and consumes a large amount of time. Furthermore, the objectivity of the classification cannot be guaranteed because of the influence of the classifier's subjectivity. To solve these problems, a Spanish poetry classification framework was designed using artificial intelligence technology, which improves the accuracy, efficiency, and objectivity of classification. First, an artificial-intelligence-driven Spanish poetry classification framework is described in detail, and is illustrated by a framework diagram to clearly represent each step in the process. The framework includes many algorithms and models, such as the Term Frequency–Inverse Document Frequency (TF_IDF), Bagging, Support Vector Machines (SVMs), Adaptive Boosting (AdaBoost), logistic regression (LR), Gradient Boosting Decision Trees (GBDT), LightGBM (LGB), eXtreme Gradient Boosting (XGBoost), and Random Forest (RF). The roles of each algorithm in the framework are clearly defined. Finally, experiments were performed for model selection, comparing the results of these algorithms.The Bagging model stood out for its high accuracy, and the experimental results showed that the proposed framework can help researchers carry out poetry research work more efficiently, accurately, and objectively.

**Keywords:** artificial intelligence; Spanish poetry classification; natural language processing; Bagging

## 1. Introduction

Over the last few decades, natural language processing techniques have developed rapidly, laying a solid foundation for researchers to analyze literary works with the help of artificial intelligence. Scientists have largely focused on text categorization [1–5] among numerous natural language processing technologies. A group of Mexican researchers proposed a semi-supervised method to classify non-English texts, such as Spanish [6]. However, most studies only focus on prose or short texts, ignoring poetic texts [7]. Based on the results of research on prose or short texts, Spanish researchers have recently started to pay attention to poems, proposing different techniques to classify Spanish poems according to their stanzas, topics, and sentiments [8–14]. These studies have provided technical support for further research on poetry categorization. Over the past few years, poetry categorization has been conducted in many different languages, including Spanish, Marathi [15], Portuguese [16], English [17], Ottoman [18], Chinese [19], Punjabi [20], and Gujarati [21]. However, poetry style classification remains an unexplored area. Regarding Spanish poetry categorization, Alvaro et al. proposed a smart method to automatically classify stanzas in Spanish poetry [9], while other researchers have provided new methods for topic modeling and sentiment analysis in poetic texts.  These studies represent successful preliminary

attempts to use artificial intelligence in the study of Spanish poetry classification, providing inspiration for the use of text mining technology for poetry style classification.

Based on the above studies, this article further proposes the use of natural language processing techniques in the study of Spanish poetry. Over hundreds of years of history, Spanish poets have created a huge number of poems, which are invaluable treasures of human art. How to preserve and study these poems has become an important issue. Modern digital technology makes it easier to save poetry and access poetry resources; therefore, scientists should find new ways to study poetry using digital technology and artificial intelligence to improve the efficiency and accuracy of research.

The assessment of poetic style has always been an important problem faced by literary researchers. Accurately distinguishing poetry styles in traditional ways means that there are extremely high requirements in terms of the reader's learning experience, their professional qualities, and their empathy. Readers need to have an in-depth understanding of the historical background of poetry and the life experience of poets before making judgments. They may also need to read a large number of similar and contrasting types of poems for analogy and comparison. These tasks require a great deal of effort and physical stamina of readers, who can waste a great deal of time in undertaking the most basic classification tasks. In order to address the shortcomings of manual poetry classification, we attempt to build a new automatic Spanish poetry style classification model using text mining techniques. This creates the possibility of improving the efficiency and accuracy of poetry style classification, and may offer Spanish poetry researchers a new study platform based on artificial intelligence.

There are almost no artificial intelligence frameworks for classifying Spanish poetry, although there is a single method designed to study it. The main contributions of this paper are as follows:

- In response to the problem of the lack of artificial intelligence frameworks for the classification of Spanish poetry, an artificial-intelligence-driven Spanish poetry classification framework is designed in detail, which greatly improves the accuracy and efficiency of classification work, compensating for the shortcomings of traditional manual poetry classification tasks.
- The proposed framework includes multiple selectable algorithms, and it can be very flexible in adding newly designed algorithms. Through model selection, the most suitable method for Spanish poetry classification can be obtained.
- Experiments based on model selection were designed. The results of the experiments showed that the Bagging model exhibited higher accuracy compared to SVMs, AdaBoost, LR, and all the other models. Applying this framework, automatic Spanish poetry style classification work can be more objective and more accurate, facilitating the study of Spanish poetry.

Section 2 introduces related work, including the study of Spanish text classification and classification methods. Section 3 describes the proposed framework, including the methods used to analyze and classify different poetry styles. Application of the Bagging model is also described in detail. In Section 4, the experimental setup and the evaluation metrics are described, and the effectiveness of the approach for Spanish poetry classification is supported by the experimental results. Section 5 summarizes the study and considers future work.

## 2. Related Works

Spanish poetry classification has been carried out by a number of researchers, who have mainly concentrated on short text classification, sentiment categorization, and stanza classification. In this section, Spanish text classification, classification methods, and poetry classification in other languages are surveyed in detail.

*2.1. Spanish Text Classification*

As a preliminary attempt to apply artificial intelligence technology to text classification tasks, Spanish researchers have conducted classification experiments on short texts on the Internet. Subsequently, the application of text classification technology has been extended to poetry. Researchers have begun to pay attention to the classification of emotions, themes, and stanzas in poetry.

In 2008, Rafael et al. proposed a semi-supervised method for text categorization [6]. They highlighted the shortcomings of manual classification and supervised classification, proposing a new model based on the SVMs algorithm. However, their research only focused on the classification of news reports, which represents a limited approach. In 2017, Eric et al. proposed a short text classifier which was also based on the SVMs algorithm [7]. However, its shortcomings were the same as that in previous research: the classifier only focuses on short texts, ignoring literary texts, and, in particular, poetry. In 2013, Linda et al. [12] reported an experiment on Spanish poetry sentiment categorization. They focused on one Spanish poet, comparing two existing popular algorithms, SVMs and Bayes. In 2018, Borja applied latent Dirichlet allocation (LDA) topic modeling to Spanish sonnets to classify topics in this area of poetry [13]. It represents a preliminary attempt to build a topic classification model for Spanish poetry, but the corpus was quite limited, as modern Spanish poetry was not taken into consideration. In 2021, Alvaro et al. carried out a study which sought to automatically classify stanzas in Spanish poetry [9]. However, the results were not satisfactory. Compared with traditional manual classification, their automatic classification model had much lower accuracy, which indicates that the technology for the automatic classification of poetry still needs to be improved. In 2020, Rafael proposed an approach to automatically classify the authorship of Spanish poetry using the Bayes algorithm [6]. His research suggested the possibility of using artificial intelligence to improve classification efficiency and accuracy. In 2021, Alberto et al. proposed a semi-supervised learning approach to automatically infer the psychological, affective, and lexico-semantic categories of Spanish sonnets [8]. Their approach achieved considerable accuracy, although the corpus focused on was still limited to classical sonnets. Other researchers have also proposed methods for using artificial intelligence to study Spanish poetry, but this research was carried out from the perspective of linguistics, such as consideration of prosody or the rhythm of the poetry [22,23]. All the aforementioned studies have provided technical and ideological inspiration that has contributed to the development of the current research.

*2.2. Classification Methods*

In previous relevant research, different algorithms have been used to build classification models. Borja attempted to build an automatic topic classification model for 5078 Spanish sonnets [13]. He compared two different latent Dirichlet allocation (LDA) models to assess which produced the best results. The first was the standard LDA algorithm, and the second, for comparison, was the Latent Feature LDA (LF-LDA). Borja hypothesised that LF-LDA could reduce LDA's dependence on context. However, the results showed that the standard LDA algorithm extracted the topics more accurately. Aiala and Luis proposed a method for emotion classification of Spanish texts [11]. They chose the SVMs classifiers to build their model, reporting an accuracy of 0.7447. There were clear shortcomings to their experimental approach. First, the corpus they chose only focused on short texts on social media, while literary texts were not paid any attention; moreover, the accuracy required further improvement. Juan-Manuel and Luis-Gil built a new Spanish literary text corpus called Literary Sentiment Sentences in Spanish (LISSS) [14]. With this new dataset, they employed different models, including Algorithm J48, Naive Bayes, Naive Bayes Multinomial, and SVMs. The accuracies obtained were 58.4%, 36.4%, 51.2%, and 49.5%. The results showed that the J48 tree algorithm was the most effective for building the model. However, the accuracy still required improvement. Linda et al. proposed a new model to classify the emotions of a Spanish poet's poetry [12]. They also employed different algorithms to compare the results. The algorithms included: Decision Trees, Naive

Bayes, SVMs, neural networks, K*, and Adaboost M1. The results showed that Decision Trees resulted in the highest accuracy, which was above 61.39%. However, the algorithm still resulted in some poetry being misclassified, which was an issue that was left for future researchers to resolve.

### 2.3. Applications of Natural Language Processing

The technology mainly used in poetry classification is natural language processing. Researchers have adopted a variety of methods to study natural language processing and have applied it to multiple research areas, such as text classification and sentiment analysis.

Zhao et al. [24] proposed a novel approach to short text classification using a sequential graph neural network (SGNN). This aims to address the challenges of capturing the sequential dependencies and semantic relationships among words in short texts, which are often ignored by traditional methods. Huang et al. [25] proposed a multitask learning framework for abuse detection and emotion classification in online social media, which uses a pretrained sentiment analysis model to derive emotion labels and to construct auxiliary tasks of emotion classification, thus avoiding a lot of manual labeling. They also proposed a decoding structure containing cross-attention to further enhance the positive effect of the auxiliary task on the primary task through the cross-attentional mechanism. The main challenge in using topic modeling for automated natural language analysis in the context of a customer support center is handling incomplete or erroneous mentions due to duplication, ambiguity, and language and pronunciation errors. Papadia et al. [26] proposed a probabilistic data transformation method to address this problem and evaluated the effectiveness of the generated solutions using quantitative performance metrics. Campos et al. [27] used a combination of web-crawling, web-scraping, and automatic text summarization with natural language processing technology to build a technology recommender system that can automate the task of keeping track of recent technologies and provide recommendations to subject matter experts. Neagu et al. [28] used a classical machine learning algorithm, specifically the Bernoulli Naive Bayes classifier with Term Frequency–Inverse Document Frequency (TF_IDF) encoding, for sentiment analysis on Romanian Twitter content, which proved to be more robust to generalization than deep-learning-based methods, and which has the advantage of fast inference times and easy retrainability. Tang et al. [29] discussed three data enhancement techniques, including synonym substitution, random insertion, and random swapping, that can improve the robustness of pretrained models, such as BERT and DistilBERT, against adversarial attacks in text categorization tasks. Zhang et al. [30] proposed a new approach based on the testkin machine model for Chinese natural language processing tasks. The learning process of this method is transparent and easy to understand compared to deep-learning-based models. Liu et al. [31] proposed a new technique for directly converting spoken textual expressions into formal written expressions using a deep learning approach for correcting texts with no obvious grammatical errors or spelling mistakes in the spoken expressions. Torres et al. [32] applied five well-known machine learning classifiers for identifying renal complications and hypertensive disorders in a clinical record that was written in Spanish. Li et al [33] proposed a categorization method based on natural language processing (NLP) techniques for analyzing construction accident report texts. The technique is based on convolutional neural networks and can automatically classify accident categories based on accident text features. Ahn et al. [34] explored five state-of-the-art morphological analyzers for Korean news articles and categorized their topics into seven categories using the Word Frequency–Anti-Document Frequency and Light Gradient Boosting Machine frameworks, with the goal of improving classification accuracy. Gu et al. [35] investigated the training processes of BERT models and demonstrated that, for domains such as biomedicine which have a large amount of unlabeled text, pretraining a language model from scratch yields more benefits than continuous pretraining of a general-purpose domain-language model.

### 2.4. Poetry Classification in Other Languages

**English:** Saif et al. [17] proposed use of TF_IDF and the rough set theory (RST) algorithm to build a classification model for English poetry. The accuracy reached 90%. Their classification work mainly focused on the topics of poetry. In 2017, Durmus Ozkan Sahin et al. proposed a model to accomplish poet detection tasks. Their results showed that the sequential minimal optimization (SMO) algorithm achieved the best result, which was above 70%.

**Gujarati:** In 2017, Bhavin and Bhargav [21] built a Gujarati poetry emotion classification model. They employed NLTK to process the language and the accuracy reached 87.62%.

**Marathi:** Deshmukh et al. [15] proposed the use of SVMs algorithms to automatically classify Marathi poetry. Their accuracy finally reached 93.54%.

**Ottoman:** Ethem et al. [18] proposed the building of an Ottoman poetry classification model. They experimented with two algorithms to attribute the authorship of poetry and to identify each poem's time period. The algorithms used included SVMs and NB. The results showed that, compared to NB, SVMs was a more accurate algorithm.

**Punjabi:** In 2017, Jasleen and Jatinderkumar [20] compared 10 algorithms to classify four different categories of Punjabi poetry. The algorithms that they employed included Adaboost, Bagging, C4.5, Hyperpipes, K-nearest neighbors, NB, PART, SVMs, Voting Feature Interval (VFI), and ZeroR. SVMs reached the highest accuracy of 58.79%.

**Chinese:** Zhu et al. [19] proposed the use of Doc2Vec and XGBoost to build an automatic Chinese poetry style classification model. The accuracy of the algorithms reached above 90%.

To summarize, most previous research has focused on sentiment classification and some specific types of topic classification. With regard to Spanish poetry classification, no study has yet built a successful classification framework for poetry styles, which is what we propose in this article.

## 3. Spanish Poetry Classification Framework Driven by Artificial Intelligence

In this section, a framework for Spanish poetry classification is designed in detail, as shown in Figure 1. Firstly, TF_IDF and Doc2Vec are applied to generate poem vectors for data preprocessing. Secondly, the dataset is divided into two parts, namely a training set and a testing set. After the preprocessing work is completed, several algorithms are used to conduct model selection.

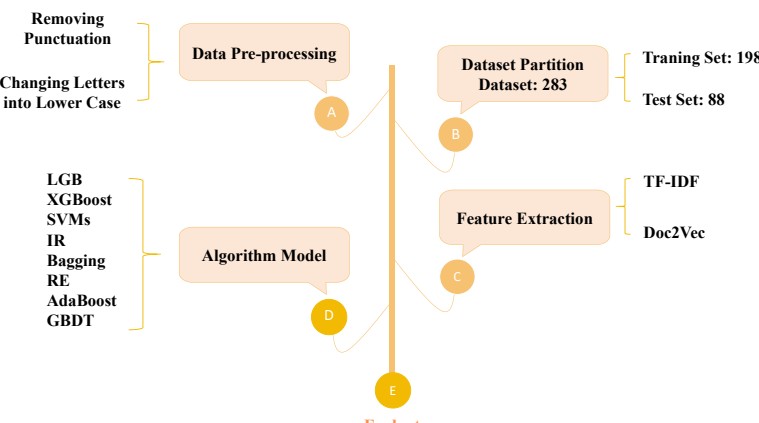

**Figure 1.** Spanish poetry classification framework. It is flexible and can include many algorithms. Through model selection, the best method suitable for Spanish poetry classification can be obtained.

*3.1. Data Preprocessing*

Classification methods are based on supervised learning techniques, so data preprocessing is of vital significance. The whole procedure of preprocessing is divided into several steps, such as labeling, removing punctuation, and changing letters to lowercase. Manually labeling the poetry is the preliminary step, which provides basic data support, ensuring that the classification is objective. Then, removing punctuation and changing letters to lowercase helps to clean the dataset, improving the accuracy.

*3.2. Feature Extraction*

To reduce the dimensions of the feature, Term Frequency-Inverse Document Frequency (TF_IDF) and Document to Vector (Doc2Vec) were employed for feature extraction.

3.2.1. Term Frequency-Inverse Document Frequency

The *TF_IDF* algorithm is a very popular tool—it mainly compares the frequency with which a word appears in a text with the number of texts. The more times that a word appears in an article, and the fewer times that it appears in all documents, the more accurately it can represent the style of the article. The *TF_IDF* algorithm is divided into two parts, *TF* and *IDF*, and these two parts are illustrated separately.

TF indicates the frequency of words appearing in the text. The formula is

$$TF_{i,j} = \frac{n_{i,j}}{\sum_k n_{k,j}}, \tag{1}$$

where $n_{i,j}$ represents the number of times that the word $t_i$ appears in the article $d_j$. This is the frequency with which the word $t_i$ appears in a document $d_j$. However, the use of this formula alone is not rigorous. Some common words do not have a large effect on the theme, but some words with a lower frequency can express the theme of the article. The choice of weights must meet the condition whereby the stronger the ability of a word to predict the subject, the greater the weight, and vice versa. In statistical articles, some words may only appear in a few articles, so such words have an important role in the topic of the article, and the weight of these words should be greater. This is enabled by *IDF*.

*IDF* indicates the prevalence of a keyword. The fewer documents that contain word $i$, the larger the *IDF* is, and the word is well-represented. The *IDF* for a particular word can be obtained by dividing the total number of articles by the number of articles containing the word, and then taking the logarithm of the resulting quotient. The formula is

$$IDF_i = \log \frac{|D|}{1 + |j : t_i \in d_j|}, \tag{2}$$

where $|D|$ represents the total number of articles, and $|j : t_i \in d_j|$ represents the number of articles containing the word $t_i$. The number 1 added before $|j : t_i \in d_j|$ serves to prevent the number of articles containing $t_i$ from being 0, resulting in an error in the operation.

To summarize, a high frequency of a word in a particular file and a low frequency of this word in the whole file set can produce a high-weight *TF_IDF*. Therefore, *TF_IDF* tends to filter out the most common words and keep the important ones. The final formula is

$$TF\_IDF = TF \cdot IDF. \tag{3}$$

3.2.2. Document to Vector

The document to vector (Doc2Vec) method is an unsupervised algorithm that learns fixed-length feature representations from variable-length text, such as sentences, paragraphs, or documents.

In Doc2Vec, each sentence of the input is represented as a unique vector. Multiple sentence vectors are combined to form a matrix D, which represents the semantic space of all sentences. Similarly, each word of the input is represented as a unique vector. Multiple

word vectors are combined to form the matrix W, which represents the embedding space of all words. A fixed length of words is sampled from a sentence at a time, taking one word as a predictor and the others as input words. Word vectors corresponding to the input word word vector and the sentence vector corresponding to the sentence paragraph vector are used as inputs in the input layer. The vectors of this sentence and the sampled word vectors are added to average or sum the results to form a new vector X, which is then used to predict the prediction words in this window.

A new sentence vector was added to the input layer in Doc2Vec. This paragraph vector or sentence vector can also be considered a word; it acts as the memory unit of the context or the topic of the paragraph. This training method is commonly called the distributed memory model of paragraph vectors (PV-DM). During training, the length of the context can be fixed and the sliding window method is used to generate the training set. Paragraph or sentence vectors are shared in this context.

After the training is completed, all the word vectors in the training sample and the corresponding sentence vector for each sentence are obtained. When predicting new sentences, the algorithm will randomly initialize the paragraph vector, insert it into the model and then iteratively obtain the final stable sentence vector according to random gradient descent. During the prediction process, softmax is used to weight the parameters from the projection layer to the output layer in the model to ensure that the vectors do not change.

We compared the results of both of these feature extraction algorithms. The results showed that the *TF_IDF* algorithm obtained the highest accuracy. The details are introduced in Section 4.

### 3.3. Algorithm Model

In accordance with the structure of the Spanish poetry classification framework, we conducted an in-depth examination of the algorithms and models employed therein. This comprehensive framework encompasses eight distinct classification algorithms and models: Light Gradient Boosting Machine (LGB), eXtreme Gradient Boosting (XGB), Support Vector Machines (SVMs), Information Retrieval (IR), Bootstrap Aggregating (Bagging), Random Forest (RF), Adaptive Boosting (AdaBoost), and Gradient Boosted Decision Trees (GBDT).

(1) Light Gradient Boosting Machine: LGBM uses gradient-boosted trees with a leaf-wise growth strategy for increased efficiency and accuracy. In the Spanish poetry framework, it offers fast training and eye-catching performance on large datasets. However, it is sensitive to noisy data and is more complex than linear models.

(2) eXtreme Gradient Boosting: XGBoost is a level-wise tree-boosting algorithm with added regularisation to prevent overfitting and enhance model performance. Effective for Spanish poetry classification, XGBoost manages missing values and capitalises on parallel processing, though training time may be longer and tuning more complicated.

(3) Support Vector Machines: SVMs is a robust classifier that constructs an optimal hyperplane for data segregation. Effective in Spanish poetry classification, it is good for high-dimensional data but performs poorly on large datasets and requires feature scaling.

(4) Information Retrieval: IR identifies key patterns and relationships in the data—a useful feature for a linguistically nuanced task like Spanish poetry classification. Its limitations remain that it is better for document retrieval, and is not purely a classification algorithm.

(5) Bootstrap Aggregating: Bagging constructs multiple decision models for decreased variance and improved model performance when classifying Spanish poetry. While it reduces overfitting, it is computationally burdensome.

(6) Random Forest: RF combines decision trees to improve predictive effectiveness. Perfect for flexibility in handling missing values and feature scaling, it is beneficial for data-rich tasks like Spanish poetry classification. However, its training speed can be slow.

(7) Adaptive Boosting: An ensemble algorithm, AdaBoost enhances classifier performance by combining weak classifiers into a strong classifier. It is useful for Spanish poetry classification, but it is sensitive to noisy data and outliers.

(8) Gradient Boosted Decision Trees: GBDT builds decision trees in a stage-wise manner to minimise residuals. Beneficial for Spanish poetry, it offers powerful predictive accuracy. However, it is slower in training, requires careful tuning and might lack interpretability compared to simpler models.

Finally, to further improve the objectivity of the framework in the classification task, we used the majority voting (MV) method to ensemble the results of eight distinct classification method. The eight classification methods in the framework are considered as eight base classifiers. By ensembling the results of multiple base classifiers, the MV method reduces the impact of individual errors and biases, which is particularly helpful for tasks such as poetry classification that are inherently subjective and ambiguous.

Assume that there are $N$ base classifiers $\{C_1, C_2, \ldots, C_N\}$ and an object $x \in X$, where $X$ is a dataset containing $M$ objects. Each classifier $C_j(x)$ outputs a predicted label for the object $x$. Then, the method of MV is defined by

$$MV(c_j, x) = \sum_{i=1}^{N} \Vdash(C_i(x) = c_j),\tag{4}$$

where $\Vdash(\cdot)$ is the indicator function that takes the value 1 when $C_i(x) = c_j$ and 0 otherwise. This means that when the classifier $C_i$ categorizes the object $x$ into the category $c_j$, the classifier will give $c_j$ a vote.

## 4. Experiment

In this section, the details of the experiments are presented, including the datasets, the evaluation method and the parameter settings.

### 4.1. Datasets

As mentioned in the Introduction, most researchers currently only focus on short texts and traditional sonnets, which are much easier to classify. To compensate for the shortcomings of these studies, a total of 283 Spanish poems were collected that were written by different poets from the 14th to the 21st centuries. These poetry examples were of three different styles: classical lyricism, modernism, and romanticism. The classification of poetic styles was derived from the opinions of authoritative scholars, which ensures that the dataset was objective and accurate. The poetry was divided into testing and training sets in order to prepare them for the experiment. Table 1 shows the training and testing sets.

**Table 1.** Distribution of the training and test sets for each poetry type in the sample data.

| The Case Category | Training Data | Testing Data |
|---|---|---|
| Classical Lyric | 67 | 32 |
| Modernism | 56 | 22 |
| Romantism | 75 | 31 |

### 4.2. Evaluation Methods

To measure and ensure the accuracy and objectivity of the algorithms, the results were evaluated from a number of different perspectives, which included accuracy, precision, recall, and the *F1_Score*. These formed part of the confusion matrix, which was especially designed for supervised learning.

(1) Accuracy: The accuracy is the total proportion of all predictions that are correct (positive and negative). The formula is

$$\text{Accuracy} = \frac{TP + TN}{TP + FN + FP + TN}.\tag{5}$$

(2) Precision: The precision is also called the accuracy rate, i.e., the proportion of correct predictions that are positive in all predictions. The formula for the precision is

$$\text{Precision} = \frac{TP}{TP + FP}.$$ (6)

(3) Recall: Recall is the proportion of what is correctly predicted to be positive that is actually positive. The formula for the recall is

$$\text{Recall} = \frac{TP}{TP + FN}.$$ (7)

(4) *F1_Score*: It is used to weigh precision and recall; generally speaking, precision and recall are negatively correlated, one is high, one is low—if both are low, there must be a problem. Generally speaking, there is a contradiction between the precision and the recall rate. The introduction of the *F1_Score* as a comprehensive index here is to balance the impact of the precision and recall rates and to evaluate a classifier more comprehensively. The *F1_Score* is a harmonic average of precision and recall. A larger value of the *F1_Score* indicates a higher quality model. It is calculated as

$$F1\_Score = \frac{2 \times \text{Precision} \times \text{Recall}}{\text{Precision} + \text{Recall}}.$$ (8)

### 4.3. Parameters Settings

In the *TF_IDF* algorithm, the parameters are set to divide the text into two distinct phrases while ignoring terms that occur in fewer than three documents and in more than ninety percent of the documents. The *TF* value is computed using a sublinear strategy. All other method parameter settings covered in this article use the default values from the original article.

### 4.4. Training and Testing of the Framework

The experiments on the Spanish poetry classification framework were designed in detail. Firstly, the dataset was preprocessed to ensure that all poems were in the same format, all letters were lowercase, and all punctuation was removed. This was the basis of all the follow-up experiments. Furthermore, the Doc2Vec and *TF_IDF* algorithms were used to extract the vectors of the features. The data were divided into a training set and test set, where the training set included 198 poems and the test set included 85 poems. Then, the task of Spanish poetry classification was performed using a variety of methods supporting the framework, including LGB, XGB, SVMs, IR, Bagging, RF, AdaBoost, GBDT, and MV. Finally, all the algorithms were compared for model selection.

### 4.5. Results Analysis

In this section, the details of the results and the data from the evaluation methods are reported.

Tables 2 and 3 show the results of the two different feature extraction algorithms. The results are also visually expressed in Figure 2.

It can be seen from Tables 2 and 3 that the average results of accuracy, precision, recall, and the *F1_Score* shown in Table 2 were better than those reported in Table 3. The average accuracy shown in Table 2 is 0.7127, while it is 0.6279 in Table 3, which indicates that TF-IDF for data preprocessing was more suitable than Doc2Vec for data preprocessing in this experiment. *TF_IDF* for data preprocessing also showed the same pattern as for accuracy for the other indicators, such as precision, recall, and the *F1_Score*, which were all better than Doc2Vec. By examination of the figure and the tables, the conclusion can be drawn that the *TF_IDF* algorithm performed better than Doc2Vec in most cases. We believe these results can be attributed to several reasons. The *TF_IDF* methodology excels in extracting textual features which are pivotal for the Spanish poetry corpus—a body of

work where style is heavily influenced by meticulous word selection and frequency. Its focus on term frequency is particularly effective in recognising the repetitive and patterned language characteristic of poetic styles. In contrast, while Doc2Vec is adept at discerning contextual and semantic nuances, the abstract and symbolic nature of poetry challenges its ability to form accurate semantic associations. Consequently, the vector representations produced by Doc2Vec may have difficulty capturing the distinctive style features inherent in different groups of poems. Figure 2 also shows that after feature extraction, Bagging achieved the best accuracy when the classification model was built.

**Table 2.** The results of the TF_IDF algorithm.

| Model/Metrices | Accuracy | Precision | Recall | *F1_Score* |
|:---:|:---:|:---:|:---:|:---:|
| Bagging | 0.8588 | 0.8923 | 0.8229 | 0.8565 |
| SVMs | 0.8353 | 0.8380 | 0.8017 | 0.8195 |
| AdaBoost | 0.7529 | 0.7616 | 0.7532 | 0.7574 |
| LR | 0.7765 | 0.8543 | 0.7311 | 0.7880 |
| LGB | 0.6076 | 0.6667 | 0.6312 | 0.6485 |
| GBDT | 0.6471 | 0.6223 | 0.6145 | 0.6184 |
| XGB | 0.6235 | 0.6250 | 0.5882 | 0.6061 |
| RF | 0.6000 | 0.5980 | 0.5765 | 0.5871 |
| MV | 0.8468 | 0.8682 | 0.7917 | 0.8282 |
| average | 0.7346 | 0.7474 | 0.7012 | 0.7235 |

**Table 3.** The results of the Doc2Vec algorithm.

| Model/Metrices | Accuracy | Precision | Recall | *F1_Score* |
|:---:|:---:|:---:|:---:|:---:|
| Bagging | 0.6471 | 0.6770 | 0.6097 | 0.6419 |
| SVMs | 0.6471 | 0.6770 | 0.6097 | 0.6419 |
| AdaBoost | 0.6000 | 0.5821 | 0.5802 | 0.5812 |
| LR | 0.5882 | 0.5905 | 0.5478 | 0.5682 |
| LGB | 0.6588 | 0.6447 | 0.6245 | 0.6345 |
| GBDT | 0.6353 | 0.6054 | 0.5986 | 0.6020 |
| XGB | 0.6588 | 0.6431 | 0.6245 | 0.6337 |
| RF | 0.5882 | 0.5359 | 0.5475 | 0.5417 |
| MV | 0.6582 | 0.6393 | 0.5775 | 0.6068 |
| average | 0.6313 | 0.6217 | 0.5911 | 0.6059 |

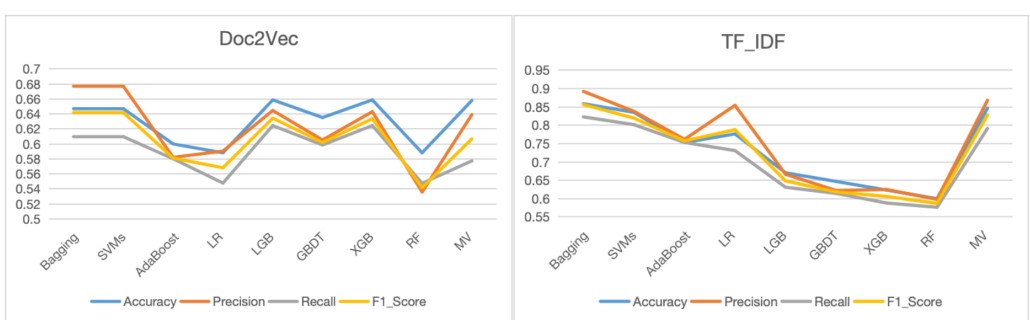

**Figure 2.** Results of the *TF_IDF* and Doc2Vec algorithms. Overall, *TF_IDF* for feature extraction obtained much better results than Doc2Vec.

## 5. Conclusions and Prospects

Poetry is an important form of literary expression and the culmination of human thought and creativity. With the continuous passage of time, new methods may be required to preserve and study poetry. Through artificial intelligence, scientists can understand and study poetry from a new perspective. In this article, the *TF_IDF* and Bagging algorithms were selected to build a classification model. The results showed that these two algorithms

had higher accuracy and objectivity than other algorithms in the Spanish poetry classification field. This demonstrates the possibility of using artificial intelligence to supplement, or even replace, manual classification, as it is more efficient and sustainable. Nonetheless, other models and algorithms need to be studied, and the database of Spanish poetry needs to be expanded, which could further improve accuracy in the automatic classification of poetry.

**Author Contributions:** Conceptualization, S.D.; methodology, S.D.; software, G.W. and S.D.; validation, S.D.; formal analysis, S.D.; investigation, S.D.; resources, F.C. and S.D.; data curation, G.W. and S.D.; writing—original draft preparation, S.D.; writing—review and editing, H.W. and S.D.; visualization, S.D.; supervision, H.W.; project administration, H.W.; funding acquisition, H.W. All authors have read and agreed to the published version of the manuscript.

**Funding:** This work was funded by the National Natural Science Foundation of China under Grant No. 62276216.

**Data Availability Statement:** Publicly available datasets were analyzed in this study. This data can be found here: https://github.com/wkwg429/SpanishPoemClassification.git (accessed on 14 October 2023).

**Acknowledgments:** My cordial thanks to Gang Wang for his efforts in researching te datas; thanks to Hongjun Wang for his careful guidance and responsible supervision. Also thanks to Fuliang Chang for his guidance in the resources of Spanish poems. This article is the result of everyone's joint efforts.

**Conflicts of Interest:** The authors declare no conflict of interest.

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
