# Peer review of "An Artificial-Intelligence-Driven Spanish Poetry Classification Framework"

_2504-2289, doi:10.3390/bdcc7040183_

Round 1

Reviewer 1 Report

Comments and Suggestions for Authors

Overall

The authors identified a novel niche and sought to fill the research gap with their contribution. The description of the relate works is rather slim, but this is more likely an indication of the lack of researchers pursuing this line of research than a superficial review. The paper is well organized. The method is clear, the results are presented well. Figures and tables are used appropriately.

Minor issue

1. Figure 1

This figure is rather confusing. Please use a standard metho of showing the pipeline as in this figure the sequence of actions is unclear. Do the data cleaning and dataset partition occur simultaneously?

Comments on the Quality of English Language

Language issues

There are many language and typographic errors throughout the paper. I only detail those in the abstract here, since more people will read the abstract than the whole paper.

1. A rashly reading of those abundant poems may confuse readers because of its complexity - -> A rash reading of those abundant poems may confuse readers because of their complexity.

2. Therefore, it’s of vital importance - -> Therefore, it is of vital importance

3. a spanish poetry classification framework - - > a Spanish poetry classification framework

4. term frequencyinverse document frequency - - > term frequency-inverse document frequency

5. the exprimental results improve that - - > the experimental results prove that

Reviewer 2 Report

Comments and Suggestions for Authors

-        Figure 1: There is a term “Data Cleaning” which in the text appears as “Data Pre-processing”. It would be easier for the reader if common terminology is adopted.

-          Section 3.2.2: The title is “Word to Vector” and the described method is “Document to Vector”. The authors should better check state of the art since Doc2Vec is an extension of the Word to Vector (Word2Vec) method.

-          Section 3.2.2 Word to Vector: There is a notion to matrix D and W without any explanation.

-          In Eq. (1) and (2) there is a “ ‘ “ in the denominator. Does it mean something?

-          Eq. (3) appears to equate the subtraction of IDF from TF (TF – IDF) with the multiplication of TF and IDF. It would be better to use a symbol like TF_IDF.

-          Eq. (5) left side: h(x)ym(x). What does it mean?

-          The algorithm performance parameters (Accuracy, Precision, Recall) are described in their form for Binary classification. However, the problem the authors describe appears to be corresponding to a multi-class classification problem ie. the algorithm output should be one of the following 3 classes: Classical Lyric, Modernism, Romanism.

-          In the results of Tables 2 and 3, the F1-score values do not satisfy the definition of F1-score as a function of Precision, Recall that the authors provide (Eq. 13).

-          In Figure 3, it would easier for the reader to read the two charts if the order of algorithms would be the same in both charts (e.g, Bagging, SVM, etc.)

-          The section 3.3 is purely state of the art material known to the readers and only a reference would be enough. Then there are some issues with what is presented. For example, bagging is not “a short of” Bootstrap Aggregating. It is another name of Bootstrap Aggregating. It is a meta-algorithm which can be applied over different underlying algorithms. The formulas provided in this section correspond to the case where the underlying algorithm is Regression. If the underlying algorithm is a Classification algorithm then Bagging works with “voting” selection criteria. In any case, this section has nothing to do with the problem the authors are addressing and provides no added value to the paper.

-          In the analysis of the results it would be beneficial to provide some insights on the reason why TF-IDF outperforms Doc2Vec method.

Comments on the Quality of English Language

The paper has a number of typos and syntax errors which have to be corrected.

Exampes include:

: "…and the exprimental results improve that the proposed…"

Use of spaces: "..algorithms:SVMs…", "…poems[22,23]…"

"…poems[13]. it’s a preliminary…"

"…model has a much less accuracy…"
“…Result showed…"

"..It aim to address..

etc.

Reviewer 3 Report

Comments and Suggestions for Authors

The manuscript proposes a machine learning framework for automatic style-based classification of Spanish poetry. In terms of topic, the manuscript is suited to the Journal. However, in terms of methodological approach and presentation, it suffers from certain drawbacks, as summarized below.

Remarks:

1. The methodological approach has been described in Section 3. However, the techniques described in this section, i.e., frequency-inverse document frequency (Section 3.2.1), word to vector (Section 3.2.2) and bootstrap aggregation (Section 3.3), have already been well-established. The actual methodological novelty of the paper remains unclear.

2. Related to the bootstrap aggregation algorithm, it has been stated that “some of the data will not be extracted, and the probability of not being extracted is 1/3” (l. 294-295). It is not clear how this probability was estimated.

3. Why h(x) on the left hand side of Eq. (5)?

4. The dataset contains (only) 283 poems grouped in (only) three style categories. This dataset is rather small and two important question remained unaddressed:
4.1 Is the dataset representative? This included two sub-questions:
4.1.1 Are the classical lyric, modernism and romanticism only relevant poem styles in the period from  the 14th to the 21th?
4.1.2 Are the poems contained in the dataset contain all stylistic phenomena related to the considered styles?
4. 2. Is the dataset balanced? I.e., is the ratio between different stylistic phenomena contained in the dataset same or at least similar to the ration between these phenomena in poems in the period from the 14th to the 21th century?
4.3. Why do the authors believe that the obtained results could be generalized?

5. When describing the F1-score (cf. l. 338-344 ), the authors incorrectly (and in several places) consider the correlation between accuracy and recall, instead of precision and recall.

6. F1-score is defined as the harmonic mean of precision and recall (as correctly stated in Eq. (13)). However, most of the F1-score values (if not all) provided in Tables 2 and 3 are not correctly calculated.

7. The conclusion that the proposed framework has higher accuracy and objectivity than other algorithms in the Spanish poetry (cf. l . 65-66  and 434-436) is not supported in the manuscript.

8. Typos (most probaly relatd to LaTex compilation):
- “ngramrange” in lines 346 and 347,
- “multiclass”, “fitintercept” and “interceptscaling” in line 368
- “maxiter“ in line 372
- additional similar typos in lines 385, 392, 398, 403-406, 409-413.
In addition:
- “Fuerthermore” in line 354.

Comments on the Quality of English Language

Please cf. Remark 8 in the review report.

Round 2

Reviewer 2 Report

Comments and Suggestions for Authors

The paper has been substantially improved since the authors have properly addressed the review comments.

Reviewer 3 Report

Comments and Suggestions for Authors

The authors have adequately addressed most of the remarks from my previous review report and I believe that the manuscript has been sufficiently improved to warrant publication.